# GBPL3 localizes to the nuclear pore complex and functionally connects the nuclear basket with the nucleoskeleton in plants

Yu Tang[1,2], Man Ip Ho[3], Byung-Ho Kang[3], Yangnan Gu[1,2]*

1 Department of Plant and Microbial Biology, University of California, Berkeley, California, United States of America, 2 Innovative Genomics Institute, University of California, Berkeley, California, United States of America, 3 School of Life Sciences, Center for Cell & Developmental Biology and State Key Laboratory of Agrobiotechnology, The Chinese University of Hong Kong, Shatin, New Territories, Hong Kong, China

* guyangnan@berkeley.edu

**Data Availability Statement:** The raw RNA-seq data reported in this study were deposited in the NCBI Gene Expression Omnibus with accession number GSE199667. The raw mass spectrometry

## Abstract

The nuclear basket (NB) is an essential structure of the nuclear pore complex (NPC) and serves as a dynamic and multifunctional platform that participates in various critical nuclear processes, including cargo transport, molecular docking, and gene expression regulation. However, the underlying molecular mechanisms are not completely understood, particularly in plants. Here, we identified a guanylate-binding protein (GBP)-like GTPase (GBPL3) as a novel NPC basket component in *Arabidopsis*. Using fluorescence and immunoelectron microscopy, we found that GBPL3 localizes to the nuclear rim and is enriched in the nuclear pore. Proximity labeling proteomics and protein-protein interaction assays revealed that GBPL3 is predominantly distributed at the NPC basket, where it physically associates with NB nucleoporins and recruits chromatin remodelers, transcription apparatus and regulators, and the RNA splicing and processing machinery, suggesting a conserved function of the NB in transcription regulation as reported in yeasts and animals. Moreover, we found that GBPL3 physically interacts with the nucleoskeleton via disordered coiled-coil regions. Simultaneous loss of *GBPL3* and one of the 4 *Arabidopsis* nucleoskeleton genes *CRWN*s led to distinct development- and stress-related phenotypes, ranging from seedling lethality to lesion development, and aberrant transcription of stress-related genes. Our results indicate that GBPL3 is a bona fide component of the plant NPC and physically and functionally connects the NB with the nucleoskeleton, which is required for the coordination of gene expression during plant development and stress responses.

## Introduction

The nuclear pore complex (NPC) is a structurally conserved multiprotein assembly embedded within the nuclear envelope and forms the main gateway to allow the nucleocytoplasmic exchange of macromolecules. The NPC is constructed by 500 to 1,000 nucleoporin proteins (Nups) of approximately 40 different types in multiple copies [1–3], which are assembled into

data reported in this study were deposited to the ProteomeXchange Database with accession number PXD032906. All other relevant data are within the paper and its Supporting Information files.

**Funding:** This work was supported by the National Institute of Food and Agriculture (HATCH project CA-B-PLB-0243-H), National Science Foundation (NSF-MCBDivision of Molecular and Cellular Biosciences 2049931), Hellman Fellows Fund, and startup funds from the Innovative Genomics Institute and University of California at Berkeley (to Y.G.). B.K. is funded by the Hong Kong Research Grant Council (GRF14121019, 14113921, AoE/M-05/12, and C4002-17G). The funders had no role in study design, data collection and analysis, decision to publish, or preparation of the manuscript.

**Competing interests:** The authors have declared that no competing interests exist.

**Abbreviations:** ABRC, Arabidopsis Biological Resource Center; BiFC, bimolecular fluorescence complementation; CC, coiled-coil; DEG, differentially expressed gene; FA, formic acid; GBP, guanylate-binding protein; GO, gene ontology; IDR, intrinsically disordered region; IRC, inner ring complex; LFQ, label-free quantitation; LFQMS, label-free quantitative mass spectrometry; LLPS, liquid–liquid phase separation; NB, nuclear basket; NE, nuclear envelope; NPC, nuclear pore complex; ORC, outer ring complex; PSM, peptide-spectrum match; mRNP, messenger ribonucleoprotein; SAGA, Spt-Ada-Gcn5-acetyltransferase; sgRNA, single guide RNA; TEAB, triethylammonium bicarbonate buffer; TF, transcription factor; TFA, trifluoroacetic acid.

4 principal NPC modules: the core scaffold, the transmembrane ring, the nuclear basket (NB), and the selective barrier [4–6]. The core scaffold forms the NPC central channel by stacking ring-shaped protein complexes, namely the inner ring complex (IRC) and outer ring complexes (ORC) that sandwich the IRC. The core scaffold makes direct contact with the transmembrane ring, which anchors the NPC to the pore membrane [7,8]. On the nuclear side, 8 flexible proteinaceous filaments project from the ORC and join at a distal ring to form the NB [9–11]. Various types of nucleoporins containing intrinsically disordered Phe-Gly (FG) repeat motifs cover the inside surface of the core scaffold and extend to the NB to form the selective barrier and enable karyopherin-mediated nuclear transport [12–15].

Among the principal NPC modules, the NB protrudes into the nucleoplasm and establishes intimate connections with inner nuclear peripheral structures. Accumulating evidence supports that the NB is a multifunctional platform that plays versatile roles beyond mediating cargo transport [16]. For example, the conserved NB scaffolding nucleoporin TPR (in animals) or Mlp1/Mlp2 (in yeasts) forms large coiled-coil (CC) homo/heterodimers that link the NPC with the underlying chromatin [17,18]. Chromatin tethering to the NB is thought to be critical for regulating transcription activity, a phenomenon known as gene gating [19,20]. In yeasts, Mlp proteins selectively recruit inducible genes by binding to their promoter regions and avoiding R-loop formation, a process that is critical for maintaining transcriptional memory and facilitating messenger ribonucleoprotein (mRNP) biogenesis [21,22]. It has been reported that a significant number of yeast transcription factors (TFs) and chromatin remodelers are able to target a chromosome region to the NPC [23], and core transcription machinery, such as the Spt-Ada-Gcn5-acetyltransferase (SAGA) transcriptional coactivator complex and the Mediator complex of RNA polymerase II, are enriched at the NB [24,25]. In metazoans, the NB-localized FG nucleoporin Nup153 regulates transcription and chromatin organization by interacting with chromatin architectural proteins and mediates their binding to *cis*-regulatory elements and topologically associating domains [26]. These and other findings collectively highlight the role of the NB in tethering the genome and regulating its activity. Moreover, TPR and Mlps are also capable of recruiting mRNPs and the TREX-2 (Transcription Elongating and RNA Export) mRNA export complex, presumably to facilitate mRNA processing and export following active transcription at the NB [27–29]. Supporting this notion, mutations in *TPR* and *Mlp1/2p* lead to nuclear retention of mis-spliced or aberrant mRNAs [30–32].

Besides interacting with the chromatin, the NB is also tightly associated with the nucleoskeleton distributed underneath the nuclear envelope. Cryo-electron tomography and super-resolution imaging analyses revealed that the NPC basket makes close contact with lamin filaments in animals [33–35]. TPR and Nup153 were reported to bind directly with lamin proteins, which is important for the proper maintenance of NPC distribution, lamin structure, and chromatin architecture [36,37].

Compared to animals and yeasts, the function of NB in plants is far less well understood. The *Arabidopsis* TPR homolog, NUA, has been shown essential for the total mRNA export. Compromised NUA function led to pleiotropic developmental phenotypes including early flowering, phyllotaxy defects, and reduced fertility [38,39]. The *Arabidopsis* FG repeats-containing nucleoporin Nup136 is likely a functional homolog of vertebrate Nup153 and has been reported to be involved in total mRNA export as well as microRNA biogenesis through its direct interaction with the TREX-2 complex [40,41]. Nup82 is a paralog of Nup136 and is plant specific. Nup82 and Nup136 interact with each other in the NB and are redundantly required for activation of SA-mediated pathogen resistance in *Arabidopsis* [42]. Nup136 and Nup82 are thought to share a common evolutionary history with a plant-specific nucleoskeleton protein KAKU4, and conserved motifs in these 3 proteins mediate physical interaction with CROWDED NUCLEIs (CRWNs), filamentous proteins that compose the nucleoskeleton

in *Arabidopsis* [43]. This finding provides an intriguing molecular mechanism for the association between the NB and the nucleoskeleton in plants. Nevertheless, the functional significance of the NB-nucleoskeleton association and how the NPC basket is involved in transcriptional regulation remain largely unknown in plants.

In this study, we describe GBPL3, a guanylate-binding protein (GBP), as an NB-associated protein in plants. GBPL3 was highly enriched at the NE and probed by multiple NPC basket and nucleoskeleton components in our previous proteomics profiling. Consistently, we showed that GBPL3 confers physical interaction with components of both the NB and the nucleoskeleton. Fluorescence imaging coupled with immunogold-labeling electron microscopy revealed that GBPL3 is predominantly located at the NPC. Proximity labeling proteomics showed that GBPL3 not only interacts with NB and nucleoskeleton proteins but also is associated with a large number of chromatin remodelers, transcription regulators, and components of RNA splicing and processing machinery, potentially facilitating transcription regulation at the NB. The *gbpl3* mutants display stunted growth and sterility. More importantly, the combination of different nucleoskeleton mutants with *gbpl3* mutant resulted in distinct development- and stress-related phenotypes and aberrant transcription of stress-related genes, suggesting a role of GBPL3 in connection with different nucleoskeleton components to differentially contribute to gene expression regulation at the NPC basket.

## Results

### GBPL3 is a candidate component of the nuclear basket/nucleoskeleton protein network

Previously, we used subtractive proteomics and proximity labeling proteomics to profile the plant nuclear envelope (NE)-associated proteome in *Arabidopsis* [44,45]. Among the identified candidates, the GUANYLATE-BINDING PROTEIN-LIKE 3 (GBPL3) protein attracted our attention. Although GBPL3 contains no predictable transmembrane (TM) domains, it was one of the top-ranked NE-associated proteins identified by the subtractive proteomics analysis (fold change = 580 and *p*-value = $1.2 \times 10^{-9}$) with a high peptide-spectrum match (PSM) score (S1A Fig), suggesting that GBPL3 is an NE protein candidate of high confidence and likely an abundant component of the plant NE. On the other hand, proximity labeling proteomics revealed that GBPL3 was probed by multiple inner nuclear membrane baits, including SUN1, MAN1, PNET2_A, and PNET2_B, by main nucleoskeleton proteins, including CRWN1 and KAKU4, and by the NPC basket nucleoporin Nup82 (Fig 1A). In contrast, outer nuclear membrane protein baits, including SINE1 and WIP1, did not probe GBPL3. This evidence suggests that GBPL3 is likely a protein associated with the inner nuclear membrane.

To investigate the potential connection between GBPL3 and nuclear envelope components, we analyzed the transcriptional correlation of *GBPL3* with genes that encode the nucleoskeleton and nucleoporins in *Arabidopsis* using ATTED-II [46]. The clustered heatmap showed that *GBPL3* is strongly coexpressed and clustered into a small group with 4 nucleoskeleton genes *CRWN1/2/3/4* and the NB gene *NUA* (Fig 1B). This result suggests a potential connection of GBPL3 with the nucleoskeleton and/or the NPC basket.

GBPL3 is conserved among eukaryotes and is an ortholog of human GBPs with 2 paralogs (GBPL1 and GBPL2) in *Arabidopsis* (S1B Fig and S1 Data). Besides the canonical N-terminal GTPase domain and the C-terminal helical domain (GBP_C) preserved in all GBPs, many plant GBPL proteins, including GBPL3, evolved a unique C-terminal extension that is annotated as a CC region but is also predicted to be intrinsically disordered (Figs 1C and S1B). Compared to ordered CC domains, disordered CC domains were found overrepresented in human proteins that function in actin filaments, cell junctions, macromolecular complexes,

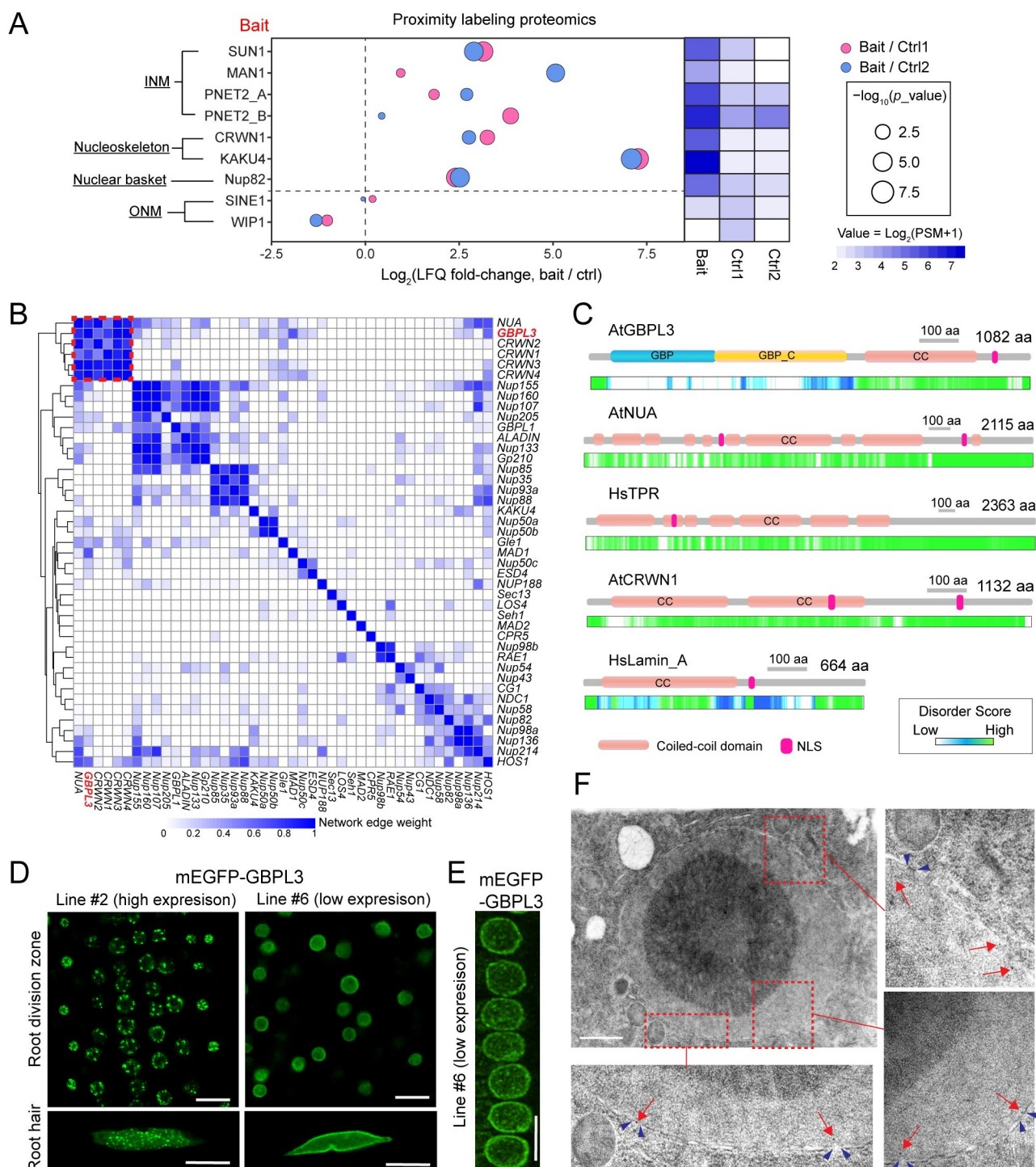

**Fig 1. GBPL3 is a nuclear envelope protein located at the NPC basket.** (A) Bubble plot showing the reanalysis of the mass spectrometry data retrieved from previously published proximity labeling proteomics using different baits (PXD015919 and PXD026924). LFQ intensity values of GBPL3 protein are plotted. A heatmap of normalized PSM values of GBPL3 are shown on the right. For bait protein SUN1, SINE1, Nup82, and WIP1, biotin-treated NEAP1-BioID2 (Ctrl 1) and mock-treated YFP-BioID2 (Ctrl 2) were used as controls. For bait protein MAN1, PNET2_A, PNET2_B, CRWN1, and KAKU4, non-transformant plants (NT) (Ctrl 1) and mock-treated YFP-BioID2 (Ctrl 2) were used as controls. (B) Weighted gene coexpression heatmap of *GBPL3* with all known genes of nucleoporin and nucleoskeleton in *Arabidopsis*. Mutual rank-based gene expression data using microarray and RNA-seq were retrieved from ATTED-II (version 11.0) and converted into network edge weights before visualizing into a clustered heatmap. (C) Prediction of CC domains and IDRs by Smart (http://smart.embl-heidelberg.de/) and D²P² (https://d2p2.pro/), respectively. (D) Fluorescence imaging showing subcellular localization of GBPL3 protein in root cells. One-week-old *pGBPL3-mEGFP-GBPL3* transgenic plants with higher expression (left

panel) and lower expression (right panel) levels are used for imaging. Bars = 10 μm. (E) Z-stack imaging followed by projection of root cells expressing mEGFP-GBPL3 in a line with lower expression. Bar = 10 μm. (F) Anti-GFP immunogold labeling and transmission electron microscopy analysis of *pGBPL3-mEGFP-GBPL3* transgenic seedlings with low expression. Root tip cells were imaged. Red arrows indicate gold particles, and blue arrowhead pairs indicate nuclear pores. Bar = 1 μm. All underlying data in Fig 1 can be found in S5 Data. CC, coiled-coil; IDR, intrinsically disordered region; INM, inner nuclear membrane; LFQ, label-free quantitation; NLS, nuclear localization signal; NPC, nuclear pore complex; ONM, outer nuclear membrane; PSM, peptide-spectrum match.

and nucleolus [47]. The disordered CC region of GBPL3 was recently demonstrated to mediate liquid–liquid phase separation (LLPS) of the protein [48]. Interestingly, part of the GBPL3 CC domain (562–784 aa) was predicted to have a structural similarity with the human Lamin_A protein (42–215 aa) by SWISS-MODEL (https://swissmodel.expasy.org) (S1C Fig), and in both *Arabidopsis* and humans, disordered CC domains are prevalent in the scaffolding NB protein (AtNUA/HsTPR) and nucleoskeleton protein (AtCRWN1/HsLamin_A) (Fig 1C), suggesting that this feature may be important for the NB and nucleoskeleton function. Together, the strong coexpression and domain similarity with NB and nucleoskeleton components led us to hypothesize that GBPL3 is functionally related to them.

## GBPL3 localizes to the NPC

To validate this hypothesis, we first verified the subcellular localization of GBPL3. We generated *Arabidopsis* stable transgenic lines expressing mEGFP-GBPL3 fusion protein. Although the construct was driven by the *GBPL3* native promoter, we were able to detect a difference in the mEGFP-GBPL3 expression level in independent transgenic lines based on immunoblots (S1D Fig), potentially due to different copy number or insertion loci of the transgene. Using confocal fluorescence microscopy, we found that in most transgenic lines with relatively low expression, mEGFP-GBPL3 is enriched at the nuclear membrane with some signal in the nucleoplasm (Fig 1D). However, in transgenic lines with high expression levels, mEGFP-GBPL3 stained the nuclear rim in a relatively weak manner but predominantly labeled bright and large droplet-like condensates at the nuclear periphery (Fig 1D), in agreement with the previous report that GBPL3 is capable of undergoing LLPS and forming biomolecular condensates upon overexpression [48]. These GBPL3 localization patterns were also observed in *mCherry-GBPL3* lines, confirming that the observation is not tag specific (S1E Fig). Because transgenic lines displaying both localization patterns can fully complement *gbpl3* mutant phenotypes (see below), we conclude that GBPL3 is functionally localized to the nuclear periphery but forming large condensates is not essential for GBPL3 function, at least under normal conditions. However, we cannot exclude that LLPS of GBPL3 may still occur at a level beyond detection by fluorescence microscope in low expression lines.

When we performed z-stack imaging of root cells using low expression lines, small punctate structures were captured at the nuclear surface (Fig 1E), similar to those formed by some plant nucleoporins (e.g., CPR5 and Nup155) [49], suggesting that GBPL3 is enriched in the NPC. To confirm this, we performed immunogold labeling and transmission electron microscopy using *mEGFP-GBPL3* transgenic lines with low expression. In nearly all samples examined, we found that gold particles are predominantly associated with the inner nuclear membrane, close to nuclear envelope openings indicative of nuclear pores (Figs 1F and S1F). Together, these results support that GBPL3 localizes at the inner nuclear membrane and is highly enriched at the NPC.

## GBPL3 physically interacts with both the nuclear basket and the nucleoskeleton

To understand the function of GBPL3 at the nuclear periphery, we performed proximity labeling proteomics using GBPL3 as bait to profile the in vivo GBPL3 proxitome. GBPL3 was fused

with TurboID, an engineered promiscuous biotin ligase that mediates efficient biotinylation of proximal proteins. The fusion construct was driven by the *GBPL3* native promoter, and we showed separately that this promoter can drive ubiquitous expression of GUS in *Arabidopsis* seedlings (except in the hypocotyl) (S2A Fig). For proximity labeling, we chose 1-week-old *pGBPL3-GBPL3-TurboID-3HA* transgenic seedlings from a line with relatively low GBPL3 expression and treated them with free biotin for 4 h followed by biotin depletion. The resulting biotinylated protein collection was affinity purified using streptavidin-coated beads. Two biological replicates were used, and wild-type non-transgenic plants were sampled in parallel as control. Immunoblot using 2% affinity-purified samples confirmed inducible and efficient protein biotinylation by GBPL3-TurboID, including GBPL3 itself (S2B Fig). Samples were then digested with trypsin on beads and purified before being subjected to label-free quantitative mass spectrometry (LFQMS), from which we collected both label-free quantitation (LFQ) intensity and PSM values. By applying fold-change > 4 and *p*-value < 0.01 as cutoffs compared to control samples, we identified 243 and 216 significantly enriched proteins using LFQ and PSM data, respectively (Fig 2A and S2 Data). We merged the 2 datasets and obtained 270 unique protein candidates, of which more than 70% are present in both datasets.

Among the significantly enriched candidates, we found all 5 known nucleoskeleton proteins in *Arabidopsis* (CRWN1, CRWN2, CRNW3, CRWN4, and KAKU4). Remarkably, GBPL3 also specifically probed all reported NB components (NUA, Nup136, Nup82, Nup50a, Nup50b, and Nup50c) but barely labeled other NPC constituents (Fig 2B). This result provides strong support that GBPL3 is part of the intimately associated NB and nucleoskeleton protein meshwork underneath the NPC. Consistently, the FG basket nucleoporin Nup82 probed GBPL3, nucleoskeleton proteins, and other basket nucleoporins (Fig 2B). However, Nup82 also probed a significant number of core scaffold nucleoporins, suggesting that compared to Nup82, GBPL3 may polarly localize to a more distal region of the NB away from the NPC core (Fig 2C).

To confirm the association between GBPL3 with the NPC basket and the nucleoskeleton, we transiently coexpressed GBPL3 with the NB nucleoporin Nup136 and Nup82 or nucleoskeleton protein CRWN1 and CRWN3 in *N. benthamiana*. Overexpression of Nup136 or Nup82 alone led to the formation of condensates at the nuclear periphery (Fig 2D), consistent with a recent report [43]. Those condensates overlapped with GBPL3 condensates (Fig 2D), suggesting that GBPL3 can form condensates at the NPC basket. On the other hand, CRWN1 and CRWN3 did not form condensates when expressed alone; however, they were recruited to GBPL3 condensates upon coexpression (Fig 2D). To determine whether GBPL3 directly interacts with Nup82/Nup136 or CRWNs, we performed yeast two-hybrid (Y2H) assays. While Nup136 displayed auto-activation as bait, we found that GBPL3 C-terminus, including the GBP_C and the disordered CC domain, is sufficient to interact with CRWN2 and weakly interact with Nup82 in yeasts (Fig 2E). The NB scaffolding nucleoporin NUA was not included in the Y2H assay because we were not able to obtain the full-length cDNA of the gene due to its large size. Lastly, we performed bimolecular fluorescence complementation (BiFC) assay and confirmed the in vivo association between the full-length GBPL3 and Nup82/Nup136/CRWNs, which occurs in condensate structures (Fig 2F). Together, these results led us to conclude that GBPL3 physically associates with both the NB and the nucleoskeleton in plants.

## GBPL3 can recruit the transcription apparatus and RNA processing machinery

Besides NB and nucleoskeleton components, gene ontology (GO) analysis revealed that GBPL3 proxitome is highly enriched in proteins that function in chromatin modification and

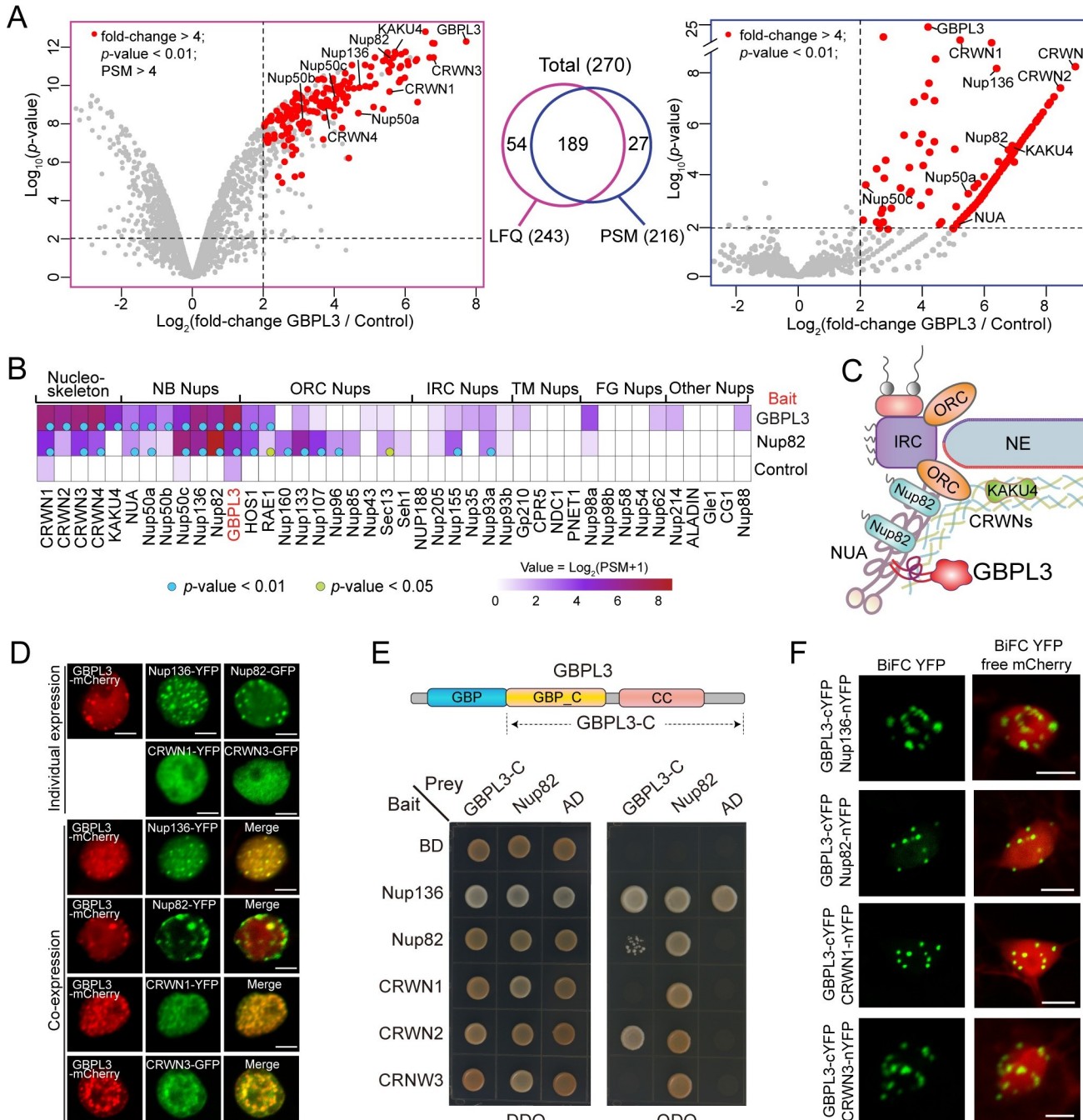

**Fig 2. GBPL3 physically interacts with NPC basket and nucleoskeleton components.** (A) Volcano plots of candidates identified by proximity labeling proteomics using GBPL3 as bait. Analyses based on the LFQ intensity and PSM data are shown on the left and right sides, respectively. The overlap of candidates identified by both analyses is shown in the middle. Non-transformants (WT) treated with biotin were used as control, and 2 biological replicates are included in the analysis. Significantly enriched proteins were selected using PSM > 4, fold-change > 4, and p-value < 0.01 as cutoffs and represented by red dots. The nuclear basket and nucleoskeleton components were text labeled. Underlying data can be found in S5 Data. (B) Heatmap showing normalized and averaged PSM values of all nucleoskeleton and nucleoporin proteins identified by GBPL3, Nup82, and KAKU4 using proximity labeling proteomics. Blue and green dots indicate p-value < 0.01 and 0.05, respectively. Nup82 and KAKU4 data were retrieved from PXD015919 and PXD026924, respectively. (C) A model diagram showing that the nuclear basket and the nucleoskeleton comprise a protein meshwork underneath the NPC and the position of GBPL3 at a more distal region of the NB. (D) Transient protein expression of GBPL3 and NPC basket components Nup136/Nup82 and nucleoskeleton components CRWN1/CRWN3 in *N. benthamiana*. The upper 2 panels show individual protein expression, and the lower 4 panels show mCherry-GBPL3 coexpression with CRWN1-YFP, CRWN3-GFP, Nup136-YFP, or Nup82-GFP. Bars = 10 μm. (E) Yeast-two-hybrid analysis using GBPL3-C (GBP_C domain and IDRs) and Nup82 as the bait and Nup136, Nup82, CRWN1, CRWN2, CRWN3 as the prey. Zygote yeasts were

grown on DDO (SD/–Leu/–Trp) and QDO (SD/–Leu/–Trp/–His/–Ade) media. Empty vectors were used as negative controls. (F) BiFC assay between GBPL3 and CRWN1, KAKU4, Nup136, or Nup82. Indicated BiFC constructs were transiently expressed in *N. benthamiana* leaf epidermal cells. Free-mCherry was coexpressed to label the nuclei. Bars = 10 μm. BiFC, bimolecular fluorescence complementation; LFQ, label-free quantitation; NB, nuclear basket; NPC, nuclear pore complex; PSM, peptide-spectrum match.

organization, transcription initiation and regulation, and RNA splicing and processing (Fig 3A), which account for over 65% of the identified GBPL3 proxitome (177/270). To better categorize the function of these proteins, we built a protein–protein interaction network of these 177 GBPL3 interactors using the STRING database, which further classified them into 14 protein complexes or functional groups with 2 main molecular functions: transcription (83 proteins) and RNA processing (94 proteins) (Fig 3B). The transcription-related category includes chromatin remodelers, histone modifiers, transcription initiator factor II (TFII) proteins, key transcription coactivators—the Mediator complex and the SAGA complex, as well as TFs and co-regulators that regulate major phytohormone responses, including auxin, gibberellin, cytokinin, salicylic acid, and jasmonic acid. The RNA processing category contains RNA helicases, mRNA splicing machinery, and other RNA processing proteins. Notably, these proteins are not enriched in the Nup82 proxitome (S3A Fig), suggesting that the GBPL3-defined NB region may be specifically enriched with molecular apparatus that is involved in active chromatin remodeling, transcription activation/repression, and RNA processing.

To confirm the association of GBPL3 with these transcription and RNA processing factors, we selected 11 well-characterized candidates, which have been reported to participate in the regulation of plant development and responses to stress-related phytohormones and various environmental stimuli [50–60]. They include chromatin remodelers ACTIN-RELATED PROTEIN 4 (ARP4), ANTHESIS PROMOTING FACTOR 1 (APRF1), and PWWP DOMAIN INTERACTOR OF POLYCOMBS1 (PWO1), transcription regulators TBP-ASSOCIATED FACTOR 5 (TAF5), *ARABIDOPSIS* RESPONSE REGULATOR 1 (ARR1), ABA-RESPONSIVE KINASE SUBSTRATE 2 (AKS2), C-TERMINAL DOMAIN PHOSPHATASE-LIKE 1 (CPL1), and TOPLESS-RELATED 1 (TPR1), and mRNA binding or processing proteins F OLIGOURIDYLATE BINDING PROTEIN 1C (UBP1C), SUPPRESSORS OF MEC-8 AND UNC-52 1 (SMU1), and SNW/SKI-INTERACTING PROTEIN (SKIP). We fused these proteins with mEGFP and transiently expressed them in *N. benthamiana*. When expressed alone, 6 of the fusion proteins formed spontaneous nuclear condensates while the other 5 diffused in the nucleus (Fig 3C). However, all of them colocalized with GBPL3-containing nuclear condensates upon coexpression with mCherry-GBPL3 (Fig 3C), suggesting that GBPL3 is capable of recruiting those chromatin remodelers, transcription regulators, and mRNA processing machinery to the nuclear periphery. These recruitment events were observed in the form of condensates with a wide range of sizes during transient overexpression (Fig 3C). Therefore, it is reasonable to speculate that GBPL3 may recruit those factors to the NB through LLPS. In line with this idea, most of these proteins contain intrinsically disordered domains predicted by multiple algorithms (S3B Fig).

Together, these results support a hypothesis that the NPC basket region in plant cells is associated with active transcription reprogramming coupled with mRNA processing, and NB-associated GBPL3 appears to be sufficient for recruiting relevant molecular apparatus for this process.

## GBPL3 is required for plant development and reproduction

To investigate the functional importance of GBPL3, we obtained 3 available T-DNA insertional mutant lines from ABRC, including SAIL_635_G05, SALK_016366 (*gbpl3-3*), and

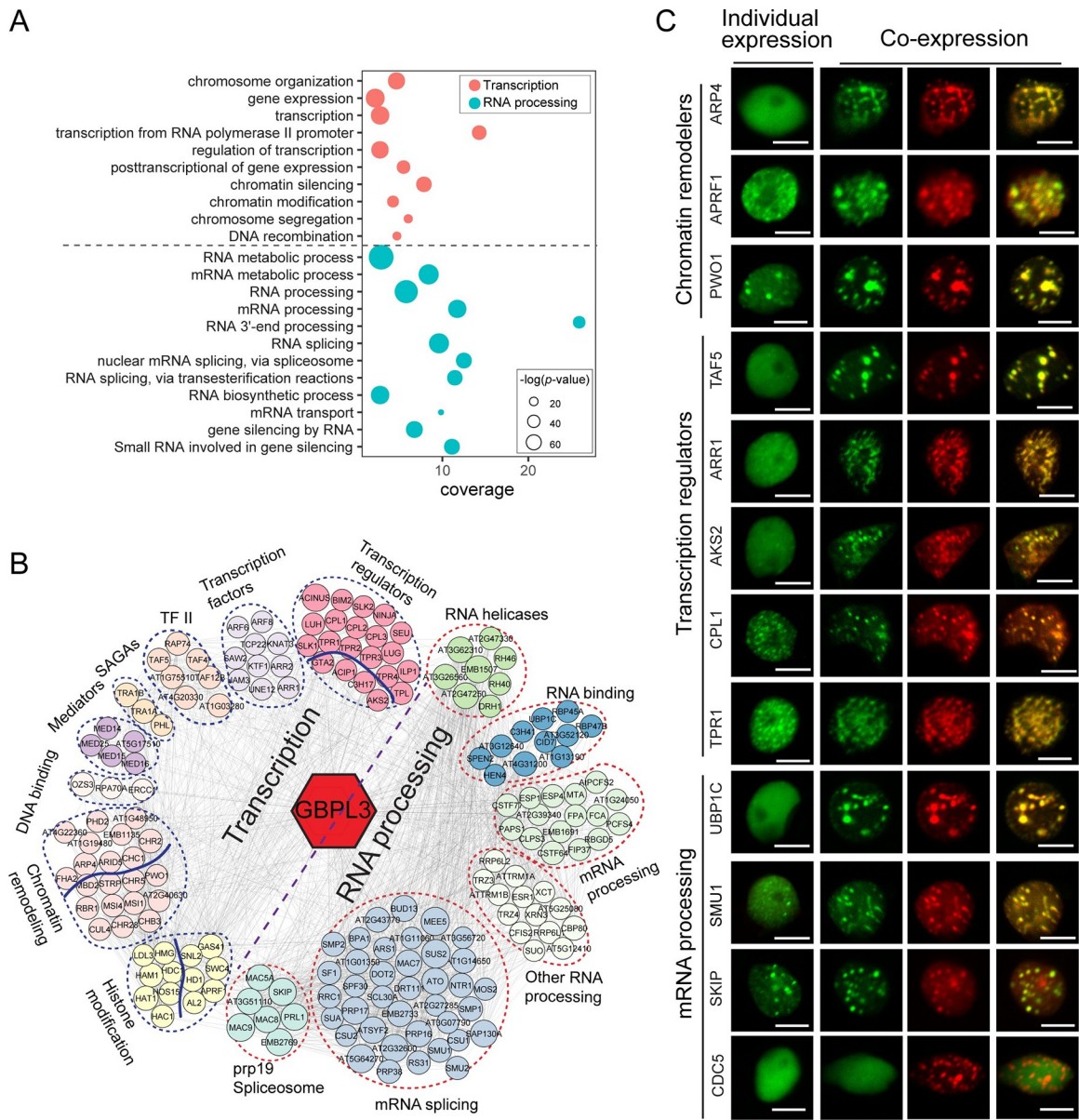

**Fig 3. GBPL3 is capable of recruiting transcription regulators and RNA processing machinery.** (A) GO enrichment analysis of GBPL3 proxitome. GO terms under the 2 major molecular functions identified (transcription and RNA processing) are presented in a bubble plot. Underlying data can be found in S5 Data. (B) GBPL3 PPI network that contains 177 GBPL3 proximal proteins involved in transcription and RNA processing identified in (A). Cytoscape was used to construct and visualize the GBP3 PPI network. GBPL3 interactors were clustered and color-coded manually based on published protein functions. Node sizes represent scores of PPIs retrieved from the STRING database (https://string-db.org/). (C) Transient expression of selected GBPL3 proximal proteins involved in transcription regulation and RNA processing in *N. benthamiana*. The left column shows the individual expression of GFP-tagged GBPL3 proximal proteins, and the right 3 columns show coexpression with mCherry-GBPL3. CELL DIVISION CYCLE (CDC5), a putative TF, was included as a negative control. Bars = 10 μm. GO, gene ontology; PPI, protein–protein interaction; TF, transcription factor.

SALK_139144 (*gbpl3-4*). However, none of the 3 alleles exhibited observable developmental defects, nor did they show decreased *GPBL3* expression measured by RT-qPCR using the whole seedling or young leaf tissue (S4A–S4C Fig). We then generated new *gbpl3* mutant alleles using CRISPR/Cas9. We designed 3 single-guide RNAs (sgRNAs) that target the coding

region of the very N-terminus of *GBPL3*, its GBP domain, and its CC domain, respectively (Fig 4A). A total of 4 different CRISPR mutant alleles were obtained, and we named them *gbpl3-6* through *gbpl3-9* (Fig 4A). In the *gbpl3-6* and *gbpl3-8* mutant, Cas9-mediated genome editing resulted in a single nucleotide insertion that led to premature termination of the GBPL3 protein (1,082 aa) at 14 aa and 696 aa, respectively. The *gbpl3-7* allele contains a 19-bp deletion that results in a premature stop codon at 116 aa. Lastly, Cas9 caused a 12-nucleotide in-frame deletion at 600 bp downstream from the start codon in the *gbpl3-9* mutant, which led to a 4 amino acid deletion in the GBP domain (S4D Fig). We found that all these 4 alleles showed developmental defects. Particularly, *gbpl3-6*, *gbpl3-7*, and *gbpl3-8* showed short roots and stunted growth (Fig 4B–4D). They are also completely sterile and are defective in pollen grain formation (Fig 4E and 4F). These phenotypes are consistent with the *GBPL3* expression pattern observed in the *pGBPL3-GUS* lines (S2A Fig) and indicate that GBPL3 is required for both normal vegetative growth and reproduction. The *gbpl3-9* mutant shows relatively weak growth defects and is semi-sterile, likely a partial loss-of-function allele (Fig 4B–4E). It has been reported that Thr75 in human GBP1 (the equivalent to Thr107 in GBPL3) is highly conserved in GTP-binding proteins and its mutation affects GTP hydrolysis but does not change GTP-binding affinity and dissociation rate [61]. These mutant alleles support that both the GTPase activity and the disordered CC region are essential for GBPL3 function.

The above phenotypes can be fully complemented by *pGBPL3-mEGFP-GBPL3* or *pGBPL3-mCherry-GBPL3* in the *gbpl3-7* mutant background (Figs 4G and S4E). Notably, this complementation is independent of the formation of GBPL3 condensates, suggesting that forming observable GBPL3 condensates is not required for plant development under normal conditions and consistent with the previous report that GBPL3-dependent biomolecular condensates are largely induced by certain biotic stresses [48].

In addition to phenotypic characterization, we also performed whole-genome RNA-seq analysis to understand the transcriptome changes in *gbpl3-7* compared to WT plants. We identified 778 up-regulated genes and 284 down-regulated genes (fold-change > 2 and *p*-value < 0.05) in 1-week-old *gbpl3-7* seedlings compared to WT (Fig 4H and S3 Data). GO term enrichment analysis suggests that both abiotic and biotic stress-related responses are spontaneously activated to certain degrees in the *gbpl3-7* mutant (Fig 4I). The mutant may also suffer from dysregulation of iron homeostasis and nutrient deficiency and other developmental defects based on the functional classification of down-regulated genes. Using RT-qPCR in the *gbpl3-6* background, we confirmed that some immune-related genes are indeed significantly up-regulated, and genes that are involved in regulating iron homeostasis and transport of nitrate are down-regulated compared to WT (S4F Fig). Together, the genetic and transcriptome analyses indicate that *GBPL3* is required for plant development and plays a role in maintaining nutrient homeostasis and suppression of stress responses under normal conditions. Notably, transcriptome responses (mainly up-regulated genes) in the *gbpl3* mutant overlapped considerably with GO terms enriched in the nucleoskeleton mutant *crwn1 crwn2* (S4G Fig and S3 Data) [53,62], suggesting a functional connection between GBPL3 and the nucleoskeleton.

## GBPL3 functions with NB and nucleoskeleton components to differentially regulate plant development and stress responses

To investigate the functional connection between GBPL3 and the NPC basket and the nucleoskeleton, we crossed *gbpl3-7* with NB nucleoporin and nucleoskeleton mutants. We found that *GBPL3* genetically interacts with the NPC basket components *Nup82* and *Nup136* in a different manner. Even though the single mutant of *nup82-2* and *nup136-3* (a weak allele) did not show observable growth phenotype, we found that *nup82-2* dramatically enhances the stunted

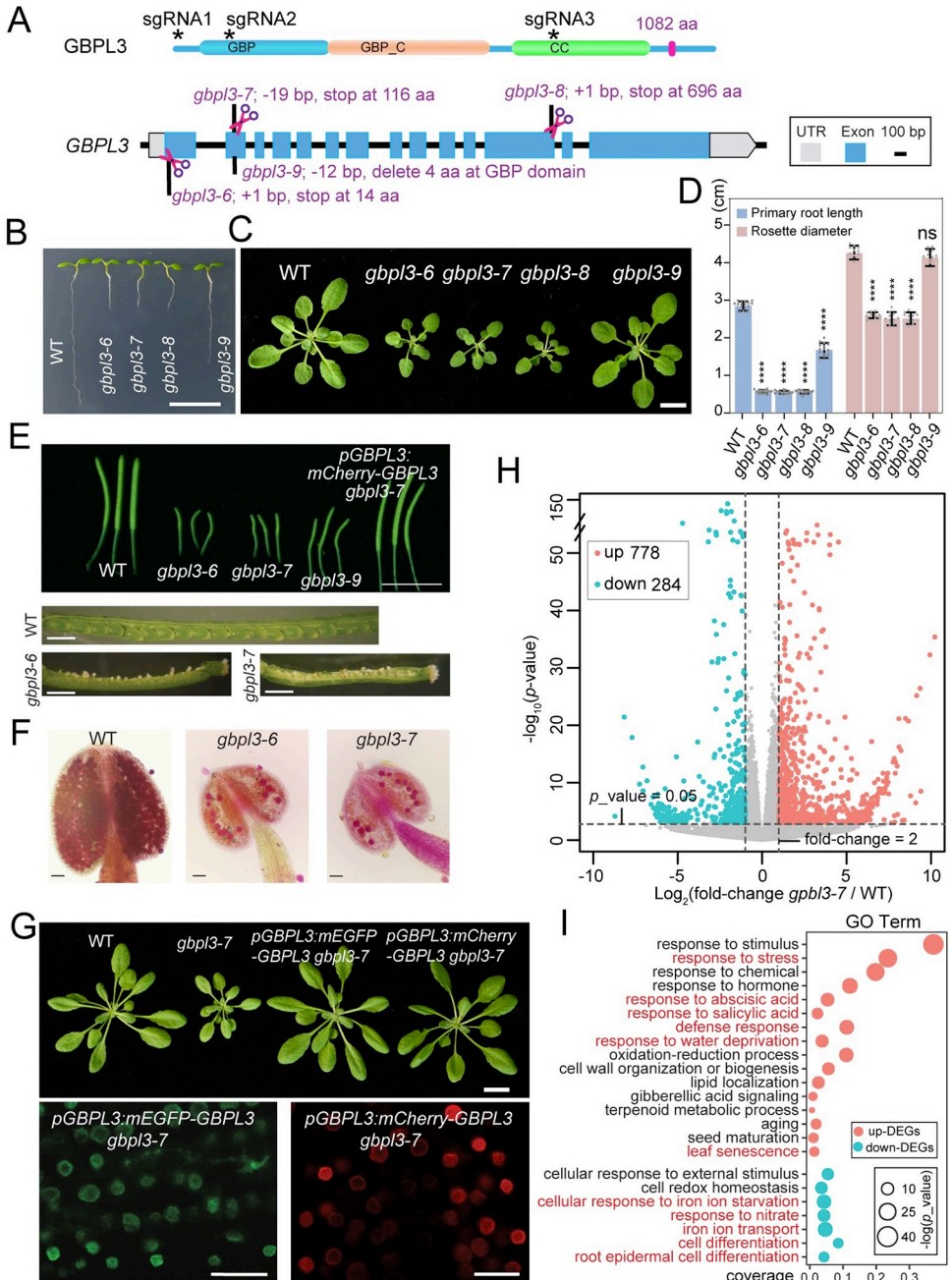

**Fig 4. GBPL3 is required for plant development.** (A) Four *gbpl3* mutant alleles generated by CRISPR/Cas9. The upper panel shows the GBPL3 protein domain structure with 3 sgRNAs that target different regions. The lower panel describes gene editing details in different *gbpl3* alleles. (B) Five-day-old WT and *gbpl3* mutant seedlings. All 4 CRISPR *gbpl3* mutants were confirmed by sequencing. Bar = 1 cm. (C) Four-week-old soil-grown plants. Bar = 1 cm. (D) Measurement of primary root length of 5-day-old seedlings and rosette diameter of 4-week-old plants ($n$ = 15 biological replicates). Student $t$ tests were performed. ****$p$-value < 0.0001; ns indicates not significant. (E) Representative siliques from 6-week-old plants. Bars = 1 cm. (F) Alexander's staining of anthers and pollen grains in WT and *gbpl3* mutants. Bars = 100 μm. (G) Complementation of the *gbpl3-7* mutant by expressing native promoter-driven *mEGFP-GBPL3* or *mCherry-GBPL3* (T3 generation). Four-week-old soil-grown plants are shown in the upper panel. Bar = 1 cm. The lower panel shows fluorescence imaging of GBPL3 localization in root cells of 5-day-old seedlings in the complemented lines. Bars = 10 μm. (H) Volcano plot showing DEGs (fold-change > 2 and $p$-value < 0.05) in the *gbpl3-7* mutant determined by RNA-seq analysis. Up- and down-regulated DEGs are displayed as pink and cyan dots, respectively. One-week-old seedlings were used ($n$ = 3 biological replicates). (I) GO term enrichment analysis of *gbpl3*-dependent DEGs. All underlying data in Fig 4 can be found in S5 Data. DEG, differentially expressed gene; GO, gene ontology; sgRNA, single guide RNA.

growth of *gbpl3-7*, whereas *nup136-3* has a very limited effect on the phenotype (Fig 5A and 5B). The *gbpl3-7 nup82-2* double mutant phenotype is further enhanced in the *gbpl3-7 nup82-2 nup136-3* triple mutant, which displays highly restricted growth and early senescence and cannot survive until reproduction. These results support a functional connection of GBPL3 with the NB and also suggest that Nup82 and Nup136 are only partially redundant, but together with GBPL3, are essential for plant survival.

Genetic interactions between *GBPL3* with nucleoskeleton genes are even more intriguing. We found that *gbpl3-7* displayed distinct phenotypes when crossed with single mutants of 4 *CRWN* paralogs, with the *gbpl3-7 crwn1* double mutant showing the most pronounced phenotype-seedling lethality (Fig 5C), suggesting a prominent/broader role of *CRWN1* among the *CRWN* family [62,63]. Both the *gbpl3-7 crwn2* and *gbpl3-7 crwn3* double mutants showed stunted growth and early senescence, which is dramatically enhanced in the *gbpl3-7 crwn2 crwn3* triple mutant, implicating a redundant role of CRWN2 and CRWN3 when acting together with GBPL3 (Fig 5D and 5E). The triple mutant plants also appear to suffer from abiotic stresses, indicated by anthocyanin accumulation. However, the *gbpl3-7 crwn2* mutant is different from the *gbpl3-7 crwn3* mutant in that the former is significantly smaller than the latter. More importantly, the *gbpl3-7 crwn3* mutant exhibited a unique trichome-less phenotype (Fig 5F), suggesting that the cell division pathway may be specifically affected in this mutant. In contrast to the above *gbpl3-7 crwn* mutants, the *gbpl3-7 crwn4* double mutant displayed no obvious phenotype compared to the *gbpl3-7* single mutant during early developmental stages (Fig 5E and 5G); however, *gbpl3-7 crwn4* plants showed chronic lesions on leaves after 5 weeks (Fig 5G), a typical phenotype caused by an elevated accumulation of salicylic acid. In line with this observation, we detected significant up-regulation of SA-responsive defense maker genes, including *PR2*, *ICS1*, and *EDS5* in the *gbpl3-7 crwn4* double mutant but not in the *gbpl3-7 crwn2* and *gbpl3-7 crwn3* double mutants (Fig 5H). Our observation of distinct *gbpl3 crwn* double mutant phenotypes raises an intriguing hypothesis that CRWNs are not redundant in their functional connection with GBPL3. We propose that GBPL3 may work with different CRWNs, which is required for distinct developmental processes and suppression of abiotic and biotic stress responses.

Taken together, these genetic results strongly support the functional connection of GBPL3 with the NPC basket and the nucleoskeleton. They also reinforce the previously established role of Nup82/136 and CRWNs in regulating plant biotic and abiotic stress responses [42,62,64] and place GBPL3 as an essential component and a critical regulator of the NB and nucleoskeleton-dependent transcription control of growth and stress in plants.

## Discussion

Emerging evidence suggests that the plant NB is intimately associated with the nucleoskeleton distributed underneath the nuclear membrane. NB nucleoporins Nup136 and Nup82 share an evolutionary origin with the nucleoskeleton protein KAKU4 and physically interact with CRWN proteins [43,64]. Here, we showed that GBPL3 interacts with both Nup136/82 and CRWNs. These extensive protein–protein interactions enable physical connections between the NB and the nucleoskeleton beneath the NPC (Fig 6). The establishment and dynamics of the joint may be contributed by liquid–liquid phase separation mediated by disordered CC region, a protein domain that can promote multivalent protein interaction. Indeed, GBPL3 was previously shown to undergo LLPS upon defense activation, and its disordered CC domain is required for this process [48]. We showed that disordered CC domains are prevalent in *Arabidopsis* NB and nucleoskeleton proteins, and GBPL3 is capable to recruit both Nup136/82 and CRWNs into condensate-like structures at the nuclear periphery upon overexpression.

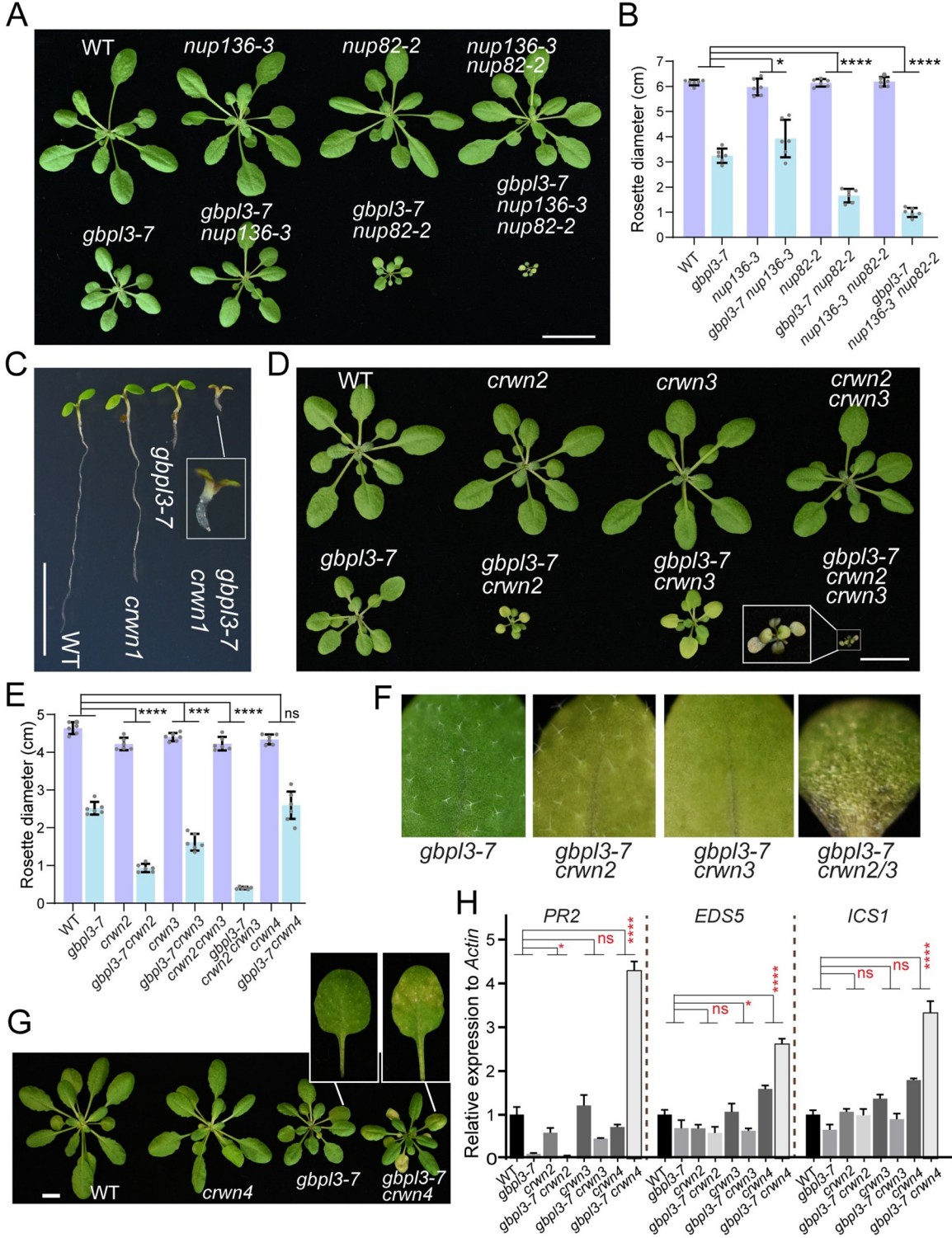

**Fig 5. GBPL3 functions with nuclear basket and nucleoskeleton components to differentially regulate plant development and stress responses.** (A) Four-week-old plants of WT, *nup136-3*, *nup82-2*, *nup136-3 nup82-2* (upper panel) and *gbpl3-7*, *gbpl3-7 nup136-3*, *gbpl3-7 nup82-2*, and *gbpl3-7 nup136-3 nup82-2* (lower panel) are shown. (B) Statistics of rosette diameter from 4-week-old plants. Bars represent means ± SD (*n* = 8 biological replicates). Similar results have been obtained 3 times. (C) Genetic interaction between *gbpl3-7* and *crwn1* mutant. Five-day-old seedlings are shown. (D) Genetic interaction between *gbpl3-7* and *crwn2/crwn3*. Four-week-old plants are shown. (E) Statistics of rosette diameter from 4-week-old plants. Bars represent means ± SD (*n* = 8 biological replicates). (F) Images and statistics of trichomes on the third rosette leaves of 4-week-old plants. (G) Four-week-old WT, *crwn4*, *gbpl3-7*, and *gbpl3-7 crwn4*

plants. Enlarged images show visible small chronic lesions on *gbpl3-7 crwn4* but not *gbpl3-7* plants. Bars = 10 μm. (H) Relative expression levels of SA-responsive defense maker gene (*PR2*, *ICS1*, *EDS5*) in WT, *gbpl3-7*, *crwn2*, *gbpl3-7 crwn2*, *crwn3*, *gbpl3-7 crwn3* and *crwn4*, *gbpl3-7 crwn4* measured by quantitative RT-PCR (*n* = 2 biological replicates). The expression level was normalized to that in WT. Two-way ANOVA was performed (*$p < 0.05$, ***$p < 0.001$, ****$p < 0.0001$). ns indicates not significant. All underlying data in Fig 5 can be found in S5 Data.

Nevertheless, the formation of visible condensates does not appear required for the role of GBPL3 in plant development. Although the evolutionary history and functional importance of the NB–nucleoskeleton interaction have not been well understood in any system, our data suggest that the interaction likely creates an integrated molecular function in facilitating gene expression coupled with RNA processing at the NPC in plant cells (Fig 6). Adding more complexity, although the 4 CRWN homologs have been shown to function redundantly in building the nucleoskeleton and in regulating chromatin organization and genome activity [62,64,65], their double mutant with *gbpl3* led to distinct phenotypes. This observation raises an intriguing hypothesis that different types of NB-nucleoskeleton joints can be formed by unique combinations between GBPL3 with CRWN proteins, which may differentially contribute to gene expression regulation. Conceivably, this differential regulation may be achieved through selective gene tethering and recruitment of different populations of TFs.

Evidence supporting a direct role of the NPC in regulating transcription activity and mRNA processing in plants is still rare. It was reported that the *Arabidopsis* ORC nucleoporin Nup85 physically interacts with a key Mediator component MED18, which is required for the expression of abscisic acid- and salt stress-responsive genes [66]. Recently, the plant-specific transmembrane nucleoporin CPR5 was identified as a novel RNA-binding protein and may work in the pre-mRNA splicing and polyadenylation complex to regulate plant immune induction [67]. We showed that the NB component GBPL3 recruits chromatin modifiers, core

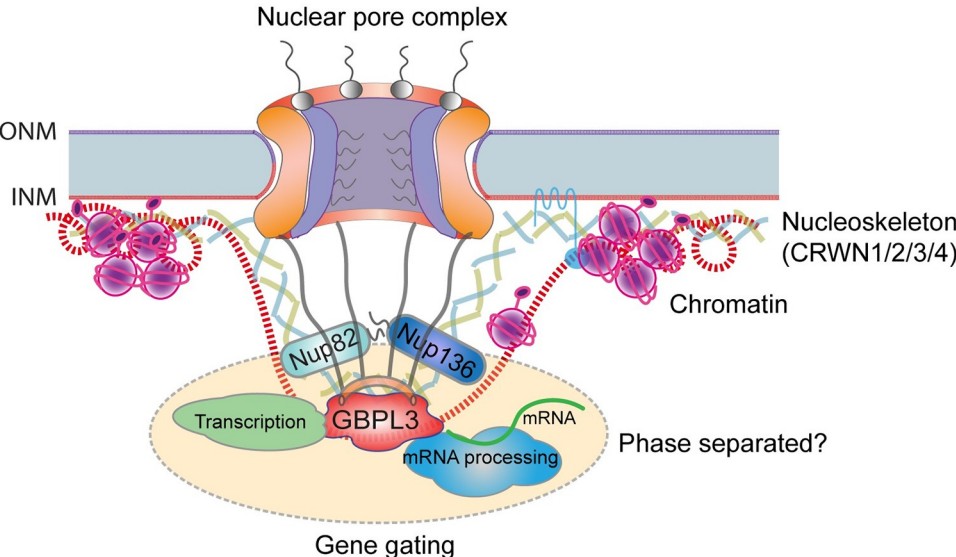

**Fig 6. Function of GBPL3 at the nuclear basket.** GBPL3 resides at a distal region of the NB away from the NPC core and associates with basket nucleoporin Nup82 and nucleoskeleton component CRWN proteins. GBPL3 may collaborate with different CRWNs to differentially regulate developmental and stress-related gene expression. GBPL3 is capable of recruiting chromatin remodelers, core transcription apparatus, TFs, and the mRNA splicing and processing machinery, and may regulate gene gating at the NPC. Molecular interactions centered at GBPL3 at the NB are possibly promoted by a phase separation process. INM, inner nuclear membrane; NB, nuclear basket; NPC, nuclear pore complex; ONM, outer nuclear membrane; TF, transcription factor.

transcription apparatus, and various TFs in vivo. These results together support an emerging view of the NPC being a hub for selective gene transcription regulation at multiple layers in plant cells, which is consistent with findings reported in animals and yeasts. Moreover, because both transcription activators and repressors are identified in the GBPL3 proxitome, the NPC basket may provide a complicated rather than simple binary (active or inactive) transcriptional environment. GBPL3 was previously shown to bind euchromatin and promoter regions of certain pathogen-related genes to regulate their expression. Pathogen infection or exogenous SA treatment could promote the formation of GBPL3 condensates, which contain specific Mediator subunits [48]. It is tempting to speculate that GBPL3 may form condensates that recruit different populations of transcription regulators under distinct stress conditions and cellular contexts, and profiling of GBPL3-associated proteins and chromatin regions would help further unravel the role of GBPL3-dependent condensates in regulating stress-induced transcription reprogramming.

Besides transcription regulation, mRNA splicing and processing factors are also recruited by GBPL3, potentially to couple mRNA maturation with active transcription at the NB. Moreover, it is known that Nup136 physically interacts with the TREX2 complex at the NB to mediate mRNA export [40,41]. These coordinated events at the NB presumably ensure rapid gene expression regulation in response to environmental and developmental cues. Interestingly, unlike Nup136, Nup82 did not probe any mRNA export/processing-related components in our proximity labeling proteomics, suggesting that Nup82 might have evolved functions that are different from Nup136 at the NB. Supporting this idea, we observed distinct phenotypes between the *gbpl3 nup82* and the *gbpl3 nup136* mutants. While *nup136* barely affects the stunted phenotype in *gbpl3*, consistent with the fact that mRNA export is downstream of gene transcription and RNA processing, the *gbpl3 nup82* double mutant displays a severely restricted growth and dramatically enhances the *gbpl3* phenotype. Although the specific function of Nup82 at the NB is still unclear, Nup136, Nup82, and GBPL3 are redundantly required for plant survival, possibly due to their common structural and/or functional connection with the nucleoskeleton.

## Materials and methods

### Plant material and growth conditions

All *Arabidopsis* plants used in this study are Col-0 background. T-DNA insertion lines, including SAIL_635_G05, *gbpl3-4* (SALK_139144), *gbpl3-3* (SALK_016366), *nup136-3* (SALK_076476), *nup82-2* (SALK_024526), were obtained from Arabidopsis Biological Resource Center (ABRC). The *gbpl3-6*, *gbpl3-7*, *gbpl3-8*, and *gbpl3-9* mutants were generated by CRISPR/Cas9-mediated gene editing. The *crwn1*, *crwn2*, *crwn3*, *crwn4*, *crwn1 crwn4*, and *crwn2 crwn3* mutant seeds were kindly provided by Dr. Eric Richard at Boyce Thompson Institute. All double or triple mutants of *gbpl3-7* with *nup82/136* or *crwns* were obtained by genetic crossing and confirmed by PCR-based genotyping. *pGBPL3*: *GBPL3-TurboID-3HA* transgenic lines are in WT background and *pGBPL3*: *mEGFP-GBPL3* and *pGBPL3*: *mCherrry-GBPL3* transgenic lines are in *gbpl3-7* mutant background. Four *pGBPL3*: *mEGFP-GBPL3* lines with lower expression were identified, and 3 lines with higher expression were identified. All transgenic plants were generated by floral dip transformation using Agrobacterium tumefaciens GV3101 and selected using Basta herbicide by spraying. All primers used for genotyping were listed in S4 Data. All seeds were surface sterilized and germinated on half-strength Murashige and Skoog (1/2 MS) agar plates after stratification at 4°C for 3 days. Seven-day-old seedlings grown on 1/2 MS medium were transferred to soil for further growth. *Arabidopsis* and *Nicotiana Benthamiana* plants were grown under a 16 h light/8 h dark-light cycle at 22°C.

## Plasmid construction

Gateway (Thermo Fisher) and In-fusion (Vazyme) technology were used for cloning unless otherwise specified. To generate constructs for protein localization, coexpression, BiFC, complementation, proximity labeling assay, *mCherry*, *mEGFP*, *n/c-YFP*, and *TurboID-3HA* were cloned into entry vector pBSDONOR p4r-2 or pBSDONOR p1-4, and cDNA fragments of *Nup136*, *Nup82* or genomic DNA fragments of *GBPL3*, *CRWN1*, *CRWN3* were cloned into entry vector pBSDONOR p1-4 or pBSDONOR p4r-2 using BP reaction (BP clonase II). The native promoter of *GBPL3* (1,314 bp upstream of the start codon) was amplified and cloned into empty vector pEG100 by In-fusion cloning (ClonExpress II One Step Cloning Kit, Vazyme). All fusion constructs were generated by combining 1 pBSDONOR p1-4 construct, 1 pBSDONOR p4r-2 construct, and an original or modified pEG100 destination vector by LR reaction (LR clonase II). To generate the construct for histochemical staining, the cDNA fragment of GUS was amplified and inserted into entry vector pBSDONOR p1-2 and further cloned into modified pEG100 containing the native promoter of *GPBL3* using LR reaction. To generate constructs for components of the GBPL3 proxitome, cDNA fragments of *ARP4*, *APRF1*, *PWO1*, *TAF5*, *ARR1*, *AKS2*, *CPL1*, *TPR1*, *UBP1C*, and *SMU1* were cloned into a modified pEG100 vector containing *GFP* by In-fusion cloning. For *PWO1*, genomic DNA fragment was cloned into pBSDONOR p1-4, which was subsequently combined with pBSDONOR p4r-2-*mEGFP* into pEG100 using LR reaction. All binary expression constructs are under 35S promoter except *GBPL3*, which is driven by its native promoter. To generate constructs for yeast-two hybrid assay, cDNA fragments of *GBPL3-C* (GBPL3 without the GTPase domain), *Nup136*, *Nup82*, *CRWN1*, *CRWN2*, *CRWN3* were subcloned into pGBKT7 or pGADT7 by In-fusion cloning. CRISPR/Cas9 constructs for gene editing were described previously [44]. Briefly, single guide RNA (sgRNA) sequences that target *GBPL3* were created using the online webserver CRISPOR (http://crispor.tefor.net/). sgRNA scaffold was then amplified using pCBC-DT1T2 vector as the template and inserted into an egg cell promoter-driven pHEE401 binary vector by golden gate assembly. All primers used for cloning were listed in S4 Data.

## Fluorescence imaging analysis

To image *Arabidopsis* plants, 5-day-old *pGBPL3-mEGFP-GBPL3* transgenic seedlings with low or high expression were used. Coexpression and BiFC assay were performed using agrobacterium-mediated transient protein expression in *N. benthamiana* as previously described [45]. Agrobacterium carrying corresponding constructs were mixed and infiltrated using a needless syringe into leaves of 4-week-old *N. benthamiana* plants, and images were taken in leaf epidermal cells 2 days past infiltration. Fluorescence images were obtained using a Zeiss LSM880 inverted confocal microscope with the GaAsP detector.

## Transmission electron microscopy and immunogold labeling

Root tips from *Arabidopsis* seedlings stably expressing mEGFP-GBPL3 were dissected and cryofixed by an HPM100 (Leica Microsystems). The frozen specimens were freeze-substituted at −80˚C and embedded in HM20 resin (Electron Microscopy Sciences) at −45˚C. After polymerization at −45˚C, samples were washed with acetone and embedded in Lowicryl HM20 resin with an increasing gradient of 33%, 66%, and 100% HM20 concentration. After UV polymerization at −45˚C, the root tips were sectioned and immunolabeled with a GFP antibody (Santa Cruz) as described previously [68]. Sample sections were observed under a transmission electron microscope after being stained with uranyl acetate and lead citrate. Gold particles in 3 individual roots of *Arabidopsis* seedlings expressing mGFP-GBPL3 and wild-type seedlings were counted. A total of 40 images were taken for each individual root, and gold particles

located on the nuclear membrane (inner nuclear membrane, outer nuclear membrane, and nuclear pore complex), nucleoplasm, and the cytoplasm were counted.

## Bioinformatic analysis

Proteomic data of GBPL3 were retrieved from our previous subtractive proteomics and proximity labeling proteomics using SUN1, MAN1, PNET2_A, PNET2_B, KAKU4, CRWN1, Nup82, SINE1, and WIP1 as baits [44,45] (PXD015919 and PXD026924). Samples were renormalized before being visualized as box plots, bubble plots, and heatmaps using R packages ggplot2 and pheatmap. To generate the co-expression network of *GBPL3* with known nucleoporin and nucleoskeleton genes, the averaged logic scores based on both microarray and RNA-seq data were retrieved from ATTED-II version 11.0 (https://atted.jp/) and visualized in a clustering heatmap using R. Protein domain prediction was performed in SMART (http://smart.embl-heidelberg.de/). Nuclear localization signals were predicted by cNLS Mapper (http://nls-mapper.iab.keio.ac.jp/cgi-bin/NLS_Mapper_form.cgi) with a score of more than 5.0. Intrinsically disordered regions (IDRs) were predicted by the Database of Disordered Protein Predictions ($D^2P^2$) (https://d2p2.pro/) and Predictor of Natural Disordered Regions (PONDR) (http://www.pondr.com/).

## Phylogenetic analysis

Phylogenetic analysis of GBPL3 was performed as previously described [44]. Briefly, GBP protein sequences from various plant and animal species including *Arabidopsis thaliana*, *Populus trichocarpa*, *Medicago_truncatula*, *Gossypium_raimondii*, *Brachypodium distachyon*, *Hordeum vulgare*, *Zea mays*, *Oryza sativa*, *Selaginella moellendorffii*, *Physcomitrella patens*, *Chlamydomonas reinhardtii*, *Caenorhabditis elegans*, *Homo_sapiens*, *Xenopus_tropicalis*, *Mus_musculus*, *Gallus gallus*, *Danio_rerio*, and, *Physcomitrium patens* were downloaded from JGI database (https://genome.jgi.doe.gov/portal/). The sequence alignment was carried out using ClustalW2, and the phylogenetic tree was built by MEGA-X (https://www.megasoftware.net/) using the maximum likelihood method with 1,000 bootstrap replicates. All sequences of GBPL3 homologs are provided in S1 Data.

## Histochemical staining with X-Gluc

To examine the expression pattern, GUS staining was performed as described previously [69]. Entire seedlings and tissues including leaf, stem, flower, anther, and silique from transgenic plants expression *pGBPL3-GUS* were used. All fresh tissues were incubated in the darkness overnight at 37°C following vacuum infiltrated in the staining buffer (100 mM sodium phosphate buffer (pH 7.0), 10 mM EDTA, 0.1% Triton X-100, 2 mM potassium ferrocyanide, 2 mM potassium ferricyanide, 1 mM X-gluc (5-bromo-4-chloro-3-indolyl glucuronide)) for 1 h. The stained seedlings or tissues were washed with 70% ethanol to remove chlorophyll before being imaged using Zeiss Lumar Stereoscope.

## Yeast two-hybrid

Yeast two-hybrid analysis was performed using the Matchmaker GAL4-based Two-Hybrid System 2 (Clontech) according to the manufacturer's instructions. The activation domain fusion (prey) constructs and the DNA-binding domain fusion (bait) constructs were transformed into yeast strains Y187 and AH109, respectively. The yeast cells of Y187 and AH109 expressing corresponding proteins were mated in 2 × YPDA medium at 30°C for 24 h before

plated and selected on double (SD-Leu-Trp) and quadruple (SD-Leu-Trp-Ade-His) dropout media at 30˚C for 3 to 5 days.

## Immunoblot analysis

Total protein was extracted from transgenic seedlings using protein extraction buffer (50 mM Tris (pH 7.5), 150 mM NaCl, 0.5% Triton X-100, 0.5% Nonidet P-40, 0.5% Nadeoxycholate, plant protease inhibitor cocktail, 1 mM PMSF, and 40 μm MG132). Protein extracts were separated by SDS-PAGE, followed by immunoblot analysis using an anti-GFP antibody (Clontech, Cat #632381, dilution 1:5,000), anti-HA antibody (Roche, Cat #11867431001, dilution 1:5,000), or streptavidin-HRP (Abcam, Cat#7403, dilution 1:10,000).

## Proximity labeling and affinity purification

The TurboID-based proximity labeling and affinity purification of biotinylated protein have been described previously with modifications [70]. Briefly, 1-week-old *pGBPL3-GBPL3-TurboID-3HA* transgenic seedlings and wild-type non-transformants were treated with 50 μm biotin for 4 h at room temperature. For each sample, 0.4 g of treated seedlings were harvested and frozen directly in liquid nitrogen. Two biological replicates were used for each sample. The material was grounded into fine powder, and total protein was extracted with protein extraction buffer (50 mM Tris (pH 7.5), 150 mM NaCl, 0.5% Triton X-100, 0.5% Nonidet P-40, 0.5% Nadeoxycholate, protease inhibitor cocktail, and 40 μm MG132). Protein extraction was subjected to 2 tandem PD-10 desalting columns (GE-Healthcare) to deplete free biotin. The following flow-through was collected for affinity purification. The eluted protein fraction was mixed with 50 μL pre-washed streptavidin-coated magnetic beads (Dynabeads MyOne Streptavidin C1, Invitrogen). The sample was incubated on a rotor wheel overnight at 4˚C, and the beads were separated on a magnetic rack and washed 5 times with the protein extraction buffer.

## On beads digestion, mass spectrometry, and proteomics analysis

For on-beads tryptic digestion, the streptavidin beads were washed with PBS buffer 3 times and incubated with 1 μg trypsin in 100 μL 50 mM triethylammonium bicarbonate buffer (TEAB) solution overnight at room temperature with gentle shaking. The resulting digests were separated from the beads on a magnetic rack and dried down by Speedvac. The peptides were dissolved with 10 μL 0.1% trifluoroacetic acid (TFA) and desalted using 10 μL C18 desalting ZipTips according to the manufacturer's instructions. The purified peptides were then dried down and re-dissolved with 20 μL formic acid (FA) before being subjected to LC-MS/MS analysis. MS/MS spectra were searched against TAIR 11 database using Scaffold 5 and MSFragger 3.2 software with default criteria to harvest PSM and LFQ intensities, respectively. To analyze proximity labeling data, both PSM and LFQ data were used as input, and DESeq2 and DEP packages in R were applied to enrich GBPL3 proximal candidates with cutoffs: fold-change $> 4$, $p < 0.01$, and normalized PSM $> 4$. Volcano plot and heatmap were generated by R. To construct the protein–protein interaction networks of GBPL3, interactions were retrieved from the STRING database (Search Tool for the Retrieval of Interacting Genes/Proteins) (https://string-db.org/) and visualized by Cytoscape (version 3.9.0).

## RNA extraction and real-time PCR

RNA was extracted using TRIzol reagent (Invitrogen, Cat #15596026) from about 0.1 g of 5-day-old seedlings. The first-strand cDNA was synthesized using the Maxima First Strand

cDNA Synthesis with dsDNase Kit (Thermo Scientific, Cat #K1672). SYBR Green PCR Master Mix (Thermo Fisher, Cat # 4309155) was used for qPCR reaction, which was performed on a CFX96TM Real-Time PCR Detection System (Bio-Rad). *ACTIN2* was used as a reference gene, and 2 to 3 biological replicates were used for each sample. All qPCR primers can be found in S4 Data.

## RNA sequencing analysis

For RNA-seq, 1-week-old media-grown Col-0 and *gbpl3-7* seedlings were collected. Total RNA was extracted using the Direct-zol RNA Miniprep kit (Genesee Scientific, Catalog #11–330) with a DNase treatment, following the manufacturer's instructions. Library construction, quality control, and RNA sequencing were performed by Novogene (https://en.novogene.com/) on an Illumina HiSeq 2000. Each sequencing run results in 38 to 50 million raw reads per sample. Sequencing read counts were quality checked and trimmed to get rid of adaptor contaminations and low-quality reads. Cleaned reads were aligned to the reference genome of *Arabidopsis* (TAIR10) using Hisat2 (v2.1.0). SAM files were sorted and converted to BAM files using SAMtools (v1.14). Read counts of each gene were generated using HTseq-count with default parameters. DESeq2 package (v3.14) was utilized to analyze the difference in gene expression. Differentially expressed genes (DEGs) were determined with criteria of $p$_value $< 0.05$ and fold change $> 2$. Gene ontology annotation of DEGs was conducted using agriGO v2.0 (http://systemsbiology.cau.edu.cn/agriGOv2/). The volcano plot and bubble plot were generated using R package ggplot2 (v3.3.5).

## Supporting information

**S1 Fig. Identification of GBPL3 as a nuclear envelope protein candidate.** (A) GBPL3 is significantly enriched in the NE fraction compared to the total MM fraction. Data were retrieved from previously published subtractive proteomics data (PXD015919). Student *t* test was performed using transformed and averaged PSM values. (B) Phylogenetic analysis of GBPL3 and its homologs from various plant and animal species. The tree and bootstrap values on each branch were generated by MEGA-X using the maximum likelihood method with 100 bootstraps. Sequences used for the phylogenetic analysis can be found in S1 Data. (C) Structural similarity prediction of GBPL3 using SWISS-MODEL (https://swissmodel.expasy.org). (D) Immunoblots with an anti-GFP antibody using total protein extract from wild-type non-transformants and 2 *pGBPL3-mEGFP-GBPL3* transgenic seedlings expressing representative higher and lower levels of GBPL3. Ten 1-week-old T3 seedlings were pooled for protein extraction. (E) Fluorescence imaging showing subcellular localization of GBPL3 protein in root cells. One-week-old *pGBPL3- mCherry-GBPL3* transgenic plants with higher expression (upper panel) and lower expression (lower panel) levels are used for imaging. Bars = 10 μm. (F) Quantification of GBPL3-GFP immunogold labeling in the low expression line (#6) and WT non-transformant samples. The relative gold density at the nuclear membrane, the nucleoplasm, and the cytoplasm is shown (left). The percentage of gold particles around the nuclear membrane was further quantified in 3 regions: the NPC, the INM, and the ONM (right). For both WT and the *GBPL3-GFP* line, 3 root samples with 20 cells and approximately 300 gold particles from each were used for the quantification. All underlying data in S1 Fig can be found in S5 Data. INM, inner nuclear membrane; MM, microsomal membrane; NE, nuclear envelope; NPC, nuclear pore complex; ONM, outer nuclear membrane; PSM, peptide-spectrum match. (PDF)

**S2 Fig. Expression pattern of *GBPL3* and proximity labeling proteomics using GBPL3 as bait.** (A) GUS staining of transgenic *Arabidopsis* plants harboring the *pGBPL3:GUS* construct. A 5-day-old seedling, the fifth and 10th rosette leaf of a 4-week-old plant, stem, and silique are shown on the upper panel from left to right, and flower and anther tissues are shown on the lower panel. Bars = 1 cm. (B) Immunoblots with HRP-conjugated streptavidin and anti-HA antibody. Wild-type and transgenic seedlings expressing GBPL3-TurboID-3HA were treated with 50 μm free biotin for 4 h before total protein extraction (Input). The total protein was then AP with streptavidin-coated beads to enrich biotinylated protein. Two biological replicates were shown. The asterisks represent naturally biotinylated proteins. AP, affinity purified. (PDF)

**S3 Fig. GBPL3 is associated with transcription regulators and RNA processing machinery.** (A) Heatmap showing normalized and averaged PSM values of transcription-related and mRNA processing-related GBPL3 proximal proteins from proximity labeling proteomics using GBPL3 and Nup82 as bait. Underlying data can be found in S5 Data. (B) Prediction of IDR in transcription-related and mRNA processing-related GBPL3 proximal proteins. IDRs predicted by $D^2P^2$ (https://d2p2.pro/) and by PONDR (http://www.pondr.com) are shown in the box and schematic diagram, respectively, for each protein. IDR, intrinsically disordered region; PSM, peptide-spectrum match. (PDF)

**S4 Fig. Loss of GBPL3 restricts plant growth and activates stress responses in *Arabidopsis*.** (A) The insertion position of the 3 *gbpl3* T-DNA lines. T-DNA insertions in SAIL_635_G05 and *gbpl3-3* were located at the 5′ UTR of *GBPL3* gene, and *gbpl3-4* mutant contains a T-DNA insertion at the 3′ UTR. Arrows indicate qPCR primers. (B) Three-week-old soil-grown WT and *gbpl3* T-DNA insertion mutant (SAIL_635_G05, *gbpl3-3*, and *gbpl3-4*) plants. (C) Relative expression level of *GBPL3* in WT and the 3 *gbpl3* T-DNA insertion mutant lines. RT-qPCR was performed using both 5-day-old whole seedlings and the 10th leaves from 4-week-old plants. Data were presented as means ± SD ($n$ = 3 biological replicates). *Actin* was used as the reference gene. Student *t* test was performed using WT as control. $^{**}p$-value $< 0.01$; ns stands for not significant. (D) PAM sequences of gRNAs and sequencing results showing CRISPR/Cas9-directed editing in *gbpl3* mutant lines. (E) Complementation of the *gbpl3-7* mutant by expressing native promoter-driven *mEGFP-GBPL3* or *mCherry-GBPL3* (T3 generation). One-week-old seedlings are shown. Bars = 1 cm. (F) Relative expression levels of immune-related and ion/nutrient-related maker gene in WT and *gbpl3-6* measured by RT-qPCR ($n$ = 2 biological replicates). Expression level was normalized to that in WT. Student *t* tests were performed ($^*p < 0.05$, $^{***}p < 0.001$, $^{****}p < 0.0001$). ns indicates not significant. (G) The overlaps of up-regulated and down-regulated DEGs of *crwn1 crwn2* (GSE106615) and *gbpl3-7* (GSE199667). The significance level of the overlap was measured using hypergeometric tests. The list of overlapping DEGs was provided in S3 Data. All underlying data in S4 Fig can be found in S5 Data. DEG, differentially expressed gene. (PDF)

**S1 Data. Protein sequences used for the phylogenetic analysis of GBPL3.** (XLSX)

**S2 Data. Proximity labeling proteomics data using GBPL3 as bait.** (XLSX)

**S3 Data. RNA-seq analysis of the *gbpl3* mutant.** (XLSX)

**S4 Data. All primers used in this study.**
(XLSX)

**S5 Data. All numerical values that underline chart summary data in figures.**
(XLSX)

**S1 Raw Images. Raw images.**
(PDF)

## Acknowledgments

We thank Dr. Eric Richards at Boyce Thompson Institute for sharing *crwn* mutant seeds and Dr. Min Jia for sharing *SKIP-GFP* and *CDC5-GFP* constructs. We also thank Dr. Denise Schichnes from the College of Natural Resources Biological Imaging Facility at UC Berkeley for assistance with fluorescence imaging.

## Author Contributions

**Conceptualization:** Yu Tang, Yangnan Gu.

**Data curation:** Yu Tang, Man Ip Ho, Byung-Ho Kang.

**Formal analysis:** Yu Tang, Man Ip Ho, Yangnan Gu.

**Funding acquisition:** Yangnan Gu.

**Investigation:** Yu Tang, Man Ip Ho.

**Methodology:** Yu Tang.

**Project administration:** Yangnan Gu.

**Resources:** Yangnan Gu.

**Supervision:** Byung-Ho Kang, Yangnan Gu.

**Validation:** Yu Tang, Byung-Ho Kang, Yangnan Gu.

**Visualization:** Yu Tang.

**Writing – original draft:** Yu Tang, Yangnan Gu.

**Writing – review & editing:** Yu Tang, Yangnan Gu.

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
