## [Editor Report · Decision Letter 0]

16 Jun 2022

Dear Dr Gu, 

Thank you for submitting your manuscript entitled "GBPL3 is a novel component of the nuclear pore complex in plants and functionally connects the nuclear basket with the nucleoskeleton" for consideration as a Research Article by PLOS Biology.

Your manuscript has now been evaluated by the PLOS Biology editorial staff as well as by an academic editor with relevant expertise and I am writing to let you know that we would like to send your submission out for external peer review.

Once your full submission is complete, your paper will undergo a series of checks in preparation for peer review. After your manuscript has passed the checks it will be sent out for review. To provide the metadata for your submission, please Login to Editorial Manager (https://www.editorialmanager.com/pbiology) within two working days, i.e. by Jun 20 2022 11:59PM.

Kind regards,

Ines

--

Ines Alvarez-Garcia, PhD

Senior Editor

PLOS Biology

---

## [Decision Letter · Decision Letter 1]

23 Aug 2022

Dear Dr Gu,

Thank you for your patience while your manuscript entitled "GBPL3 is a novel component of the nuclear pore complex in plants and functionally connects the nuclear basket with the nucleoskeleton" was peer-reviewed at PLOS Biology, and please accept my apologies for the delay in sending you our decision, mainly due to the holiday period. It has now been evaluated by the PLOS Biology editors, an Academic Editor with relevant expertise, and by three independent reviewers.

The reviews are attached below. As you will see, the three reviewers are all positive and only ask for minor changes. Based on the reviews, we are likely to accept this manuscript for publication, provided you satisfactorily address the remaining points raised by the reviewers. Please also make sure to address the data and other policy-related requests stated below.

Please note that one of the reviewers requests a change of title, thus we would like you to consider the following suggestion:

"GBPL3 localizes to nuclear pore complex in plants and functionally connects the nuclear basket with the nucleoskeleton"

We expect to receive your revised manuscript within two weeks. 

*Published Peer Review History*

*Press*

Sincerely,

Ines

Ines Alvarez-Garcia, PhD

Senior Editor

PLOS Biology

DATA POLICY: IMPORTANT - PLEASE READ CAREFULLY

Fig. 1A, B; Fig. 2A; Fig. 3A; Fig. 4D, H, I; Fig. 5B, E, H; Fig. S1A, E; Fig. S3A and Fig. S4C, F

**In addition, you should make now publicly available the data you have deposited in the in ProteomeXchange database (ID: PXD032906).

We require the original, uncropped and minimally adjusted images supporting all blot and gel results reported in an article's figures or Supporting Information files. We will require these files before a manuscript can be accepted so please prepare and upload them now. Please carefully read our guidelines for how to prepare and upload this data: https://journals.plos.org/plosbiology/s/figures#loc-blot-and-gel-reporting-requirements

BLURB

Please also provide a blurb which (if accepted) will be included in our weekly and monthly Electronic Table of Contents, sent out to readers of PLOS Biology, and may be used to promote your article in social media. The blurb should be about 30-40 words long and is subject to editorial changes. It should, without exaggeration, entice people to read your manuscript. It should not be redundant with the title and should not contain acronyms or abbreviations. For examples, view our author guidelines: https://journals.plos.org/plosbiology/s/revising-your-manuscript#loc-blurb

Reviewers’ comments

Rev. 1:

This works builds from the author's previous work (Tang et al, 2020; Tang et al, 2022), identifying components of the Arabidopsis nuclear pore complex by proteomics. In these studies, the protein GBPL3 was discovered; characterization of this newly discovered component of the Arabidopsis nuclear pore complex is the subject of this current manuscript. GBPL3 recruits chromatin remodelers to the NPC, suggesting a role for transcriptional regulation. GBPL3 further interacts with the nucleoskeleton. This work is in contrast with a recent publication (Huang et al., 2021) suggesting that GBPL3 acts as a transcriptional regulator of immune response within the nucleus.

Overall, I found the data to be compelling, claims appropriately backed with data, and I particularly appreciate the caveats and alternative explanations provided by the authors. Other than the two typos listed below, I have no concerns about the data presented in this manuscript and feel that this makes a valuable addition to our understanding of the plant nucleopore and further provides fodder for future studies.

Line 53 - "In metazoan" should be "In metazoans".

Line 76 - "CROWED" should be "CROWDED".

Rev. 2:

This is a detailed and carefully executed study which significantly enhances knowledge of the plant nuclear pore complex, by adding a new and functionally significant protein.

15 However, the functional mechanisms behind- consider rewording; 'behind' is unclear as to meaning

18 we found that GBPL3 is localizes to the nuclear rim and is enriched in the nuclear pore.

26 bona fide - Latin; italicise.

74 Nup136 and Nup82 are thought to share a common evolutionary history with a plant75 specific nucleoskeleton protein KAKU4, and conserved motifs in these three proteins mediate physical 76 interaction with CROWED NUCLEIs (CRWNs), filamentous proteins that compose the nucleoskeleton in 77 Arabidopsis (39). Please provide a reference for the common evolutionary history of KAKU4.

227 formed spontaneous nuclear condensates while the rest remaining five diffuse in the nucleus

231 under the transient overexpression condition better: in transient overexpression

336- Adding more complexity, 337 although the four CRWN homologs have been shown to function redundantly in building the 338 nucleoskeleton and in regulating chromatin organization and genome activity (58-60), their double mutant 339 with gbpl3 led to distinct phenotypes, raising an intriguing hypothesis that different types of NB 340 nucleoskeleton joints can be formed by unique combinations between GBPL3 with CRWN proteins, which 341 may differentially contribute to gene expression regulation. Consider rewording the above to simplify for the reader.

Figure 1 E) Three-dimensional reconstruction of root cells expressing mEGFP-GBPL3 in a line with lower expression. Please reword for greater clarity; this is not a three dimensional reconstruction…

Rev. 3:

The manuscript from Yu Tang, Man Ip Ho, Byung-Ho Kang and Yangnan Gu entitled "GBPL3 is a novel component of the nuclear pore complex in plants and functionally connects the nuclear basket with the nucleoskeleton" is an original contribution deciphering the plant NPC structure and function. Starting from the innovative proximity labeling proteomics technique recently developed in plants by the authors, the authors here focused on a specific component called GBPL3 identified in their initial experiments on the nuclear envelope. They clearly show using bio-informatics, molecular biology techniques and genetics that GBPL3 is enriched at the NPC basket, physically interacts with lamina and NUP components, is connected to the transcription machinery and that its mutation affects plant development in an additive/synergistic genetic effect with components of the plant lamina. This manuscript provides a strong validation of the previous observations made by proximity labeling proteomics and describes a new component connected to the NPC basket. For these reasons, I strongly recommend to consider this manuscript for publication in PLOS Biology with the following minor revisions.

Minor revisions:

Title: Although there is no doubt about the fact that GBPL3 is located at the NPC basket and interacts with the lamina and NPC, it is still questionable whether it is a true component of the nuclear pore basket or rather a protein interacting preferentially with certain components of the NPC basket and connecting the NPC to the transcription machinery. I would suggest to slightly change the title.

l31: is the NPC the only possibility for a protein to access the nucleus as the ER that is in continuity to the NE can provide a way to TM anchored proteins to be imported into the nucleus.

l32: This number seems a little too large as to my knowledge the NPC is made an octomeric structure with each monomer containing about 30-40 NUPs (i.e. 240-320 proteins / NPC?)

l37-38: I recommend to add a citation describing the Plant NPC. Fiserova and Golberg 2009 used electron microcopy to investigate the NPC structure and gave some insights about the NPC basket. They proposed that filaments at the NPC basket are about 10nm.

l47: I suggest to refer to "gene gating" by citing the initial Blobel's paper (1985) or to a review such as Tamura 2020, Journal of Plant Research

l73: Can the author reformulate these two sentences? NUP82 and NUP136 are two paralogs that contain FG-repeats and are located at the NB. NUP82 and NUP136 are two plant specific proteins that are redundantly required for...

l89: Can the authors better connect this sentence with the following one?

l107-Fig1A: can the authors specify in the Fig1A legend which control is control 1 and control2.

Nup93a-BoiID2 (control1?) and YFP-BioID2 (control2?)

mock-treated samples were used as controlsYFP-BioID2 mock-treated samples (control1?) and non-transformant plants (NT) (control2?) were used as controls.

l123: can this be shown in a supplemental figure?

l134-135: The authors described a phenomenon that is likely due to the genomic position of the transgene insertion affecting the level of expression and/or that can be linked to a missing regulatory element in their construct. I feel that this section would need some editing to better explain the observed results.

First, the authors proposed that there are two set of lines (i.e. high vs low expression) among their transgeneic lines. Can they provide the number of lines produced and the number of high and low expressing lines in Fig S1C.

Second, "Even though"... I guess the authors mean that this kind of position effects are usually observed with a p35S promoter and not/less with an endogenous promoter? can this be reformulated to help the reader to understand this unexpected result?

l159: "in vivo" in italic

l188-189-Fig 2B: It is surprising that the images showing CRWN1 and CRWN3 show that they are not enriched at the NE as described in other studies. As this has already been described in other studies (see Dittmer et al 2007) I would suggest to remove the images of CRWN1 and CRWN3 expression. If possible, it would also be better to select sections of confocal images showing expression at the NE (i.e. a cross section of the median part of the nucleus).

l193-Fig 2D-E-F: For clarity; can the author provide results with CRWN2 in their transient expression assays (even if the result is negative)? CRWN4 is missing in these experiments (can the authors give an explanation?).

l197-Fig 2F: Here again, it would be also better to select sections of confocal images showing expression at the NE (i.e. a cross section of the median part of the nucleus).

l226-230: As GBPL3 form aggregates, one possibility is that it will have the ability to spontaneously attract other co-expressed proteins. I would strongly recommend to add at least one negative control in this experiment by choosing a nucleoplasmic protein that was not suggested to interact or to be in close proximity to GBPL3.

l278-281: can the author provide as a supplementary fig a venn diagram showing the overlapping between gbpl3 and crwn1 crwn2 DEGs?

can they also provide a ref to crwn1 crwn2 transcriptomic experiments (i.e. Mukulski et al 2019 and/or Choi et al 2019, ...).

l298: CRWN1 was often associated with stronger phenotypes in earlier studies from the Eric Richard's group. These previous observations can be referred here.

l311-314: Maybe the authors can also recall here that in Y2H, a clear physical interaction was recorded only with CRWN2?

l350: « in vivo » in italic

---

## [Editor Report · Decision Letter 2]

14 Sep 2022

Dear Dr Gu,

Thank you for the submission of your revised Research Article entitled "GBPL3 localizes to the nuclear pore complex and functionally connects the nuclear basket with the nucleoskeleton in plants" for publication in PLOS Biology. On behalf of my colleagues and the Academic Editor, Mark Estelle, I am happy to say that we can in principle accept your manuscript for publication, provided you address any remaining formatting and reporting issues. These will be detailed in an email you should receive within 2-3 business days from our colleagues in the journal operations team; no action is required from you until then. Please note that we will not be able to formally accept your manuscript and schedule it for publication until you have completed any requested changes.

PRESS

Sincerely, 

Ines

--

Ines Alvarez-Garcia, PhD

Senior Editor

PLOS Biology
